# Type 1 Diabetes and the Menstrual Cycle: Where/How Does Exercise Fit in?

**DOI:** 10.3390/ijerph20042772

**Published:** 2023-02-04

**Authors:** Saru Toor, Jane E. Yardley, Zeinab Momeni

**Affiliations:** 1Physical Activity and Diabetes Laboratory, Alberta Diabetes Institute, Edmonton, AB T6G 2E1, Canada; 2Immunology and Infection Program, Department of Biological Sciences, University of Alberta, Edmonton, AB T6G 2E9, Canada; 3Augustana Faculty, University of Alberta, Camrose, AB T4V 2R3, Canada; 4Faculty of Kinesiology, Sport, and Recreation, University of Alberta, Edmonton, AB T6G 2H9, Canada; 5Women’s and Children’s Health Research Institute, University of Alberta, Edmonton, AB T6G 1C9, Canada

**Keywords:** type 1 diabetes, menstrual cycle, exercise, glucose, insulin

## Abstract

Regular exercise is associated with substantial health benefits for individuals with type 1 diabetes (T1D). However, the fear of hypoglycemia (low blood glucose) due to activity-induced declines in blood glucose levels acts as a major barrier to partaking in exercise in this population. For females with T1D, hormonal fluctuations during the menstrual cycle and their effects on blood glucose levels can act as an additional barrier. The impact that these cyclic changes may have on blood glucose and insulin needs and the consequent risk of hypoglycemia during or after exercise are still unknown in this population. Therefore, in this narrative review, we gathered existing knowledge about the menstrual cycle in T1D and the effects of different cyclic phases on substrate metabolism and glucose response to exercise in females with T1D to increase knowledge and understanding around exercise in this underrepresented population. This increased knowledge in such an understudied area can help to better inform exercise guidelines for females with T1D. It can also play an important role in eliminating a significant barrier to exercise in this population, which has the potential to increase activity, improve mental health and quality of life, and decrease the risk of diabetes-related complications.

## 1. Introduction

Female hormones and their fluctuations during the menstrual cycle can affect substrate metabolism. Additionally, for females with type 1 diabetes (T1D), these fluctuations during the menstrual cycle can affect blood glucose levels and/or insulin sensitivity. The impact that these cyclic changes may have on blood glucose levels and insulin needs and the consequent risk of hypoglycemia during or after exercise are still unknown in this population. This lack of knowledge may act as an important barrier to partaking in exercise in this population, despite the many health benefits that exercise and physical activity can bring. Therefore, in this narrative review, we aimed to gather the existing information about the menstrual cycle in T1D and the effects of different cyclic phases on substrate metabolism to discuss their potential effects on glucose response to exercise in females with T1D, with the hope of increasing knowledge and understanding around exercise in this population. This increased knowledge may have the potential to not only play an important role in eliminating a significant barrier to exercise but also help to better inform exercise guidelines for females with T1D.

In identifying sources for this narrative review, multiple databases including Google Scholar, PubMed, ScienceDirect, and the University of Alberta Library search database were used. We also checked the reference lists of the articles we found in order to identify other potentially relevant publications.

## 2. Menstrual Cycle

The menstrual cycle is a monthly reproductive cycle regulated by various hormonal concentrations that prepare the female body for pregnancy [1] (Figure 1). The cycle, which on average occurs over the span of 28 days, can be divided into three main phases: the follicular, the ovulatory, and the luteal phase [2]. The follicular phase starts at the beginning of menses and lasts until ovulation [2]. During the early follicular phase, estrogen, progesterone, and luteinizing hormone (LH) concentrations are low, while the follicle stimulating hormone (FSH) level is slightly higher than the previous cycle in order to stimulate the development of the ovarian follicles [3]. During the mid to late follicular phase, estrogen level rises in parallel with the growth of the dominant follicle and reaches its peak prior to ovulation [3]. Ovulation, which occurs around the 14th day of a 28-day menstrual cycle, is stimulated by the surge in LH and FSH levels when the dominant follicle is ruptured and releases the ovum (egg) from the ovary into the fallopian tube [3].

Following ovulation, the luteal phase, i.e., the last half of the cycle, begins [3]. This phase involves the conversion of the ruptured dominant follicle into the corpus luteum. The corpus luteum secretes progesterone and estrogen during the early to mid-luteal phase, causing the progesterone level to reach its peak and estrogen concentration to remain elevated. In the absence of implantation, the corpus luteum degenerates during the late luteal phase, causing estrogen and progesterone levels to drop and menstruation to start [3].

## 3. Female Hormones and Their Function

### 3.1. Estrogen

Estrogen is a sex steroid hormone that, in females, is synthesized predominantly in the ovaries [4]. It plays a key role in inducing both ovulation and endometrium growth to prepare for implantation [5]. High estrogen concentration prior to ovulation indirectly induces a surge in LH, causing ovulation to occur [5]. Following ovulation, estrogen begins to decline but remains at elevated levels to promote the growth of the endometrium [5]. In addition to its role in regulating female reproductive functions, estrogen plays roles outside of the female reproductive system. For example, estrogen is a key regulator of bone turnover [6]. Within the cardiovascular system, it mediates cardioprotection by increasing angiogenesis, nitric oxide production and vasodilation, while decreasing reactive oxygen species, oxidative stress, and fibrosis [7,8]. Estradiol, the most potent and well-known type of estrogen [6], helps maintain the integrity of the vasculature by protecting the endothelial cells from injury and apoptosis [8]. Within the central nervous system, estrogen may play neurotrophic, neuroprotective, and psychoprotective roles due to its effects on learning, memory, mental state, and mood, along with neurodevelopmental and neurodegenerative processes [9].

Besides these effects, estrogen affects metabolic pathways [10] by upregulating both glucose uptake and insulin secretion, thereby improving glucose metabolism [11]. It is suggested that GPER (G Protein-Coupled Estrogen Receptor GPR30) activation of the epidermal growth factor receptor and ERK (extracellular signal-regulated kinase) in response to estradiol may play an important role in the secretion of insulin from beta cells [12]. In particular, estradiol protects the functionality of the pancreatic beta cells and prevents apoptosis while lowering insulin resistance and increasing insulin sensitivity [13]. Estradiol’s effects on insulin sensitivity are thought to be mediated through the inhibition of the transcription factor Foxo1 via activation of the ERα-PI3K-Akt (estrogen receptor alpha-phosphoinositide 3 kinases-protein kinase B) signaling pathway [14]. Estrogen also inhibits the gluconeogenesis pathways, leading to lower fasting glucose levels in individuals with higher estrogen levels compared to those with lower estrogen levels [14]. Despite enhancing glucose metabolism, estrogen favours lipid oxidation when lipids are available at higher concentrations than carbohydrates [13]. Most animal studies also suggest that estrogen leads to a shift in fuel oxidation from carbohydrates toward fat by stimulating lipolysis [15]. Interestingly, a study involving estrogen supplementation significantly increased lipid oxidation and decreased carbohydrate oxidation at rest and during endurance exercise in healthy males [16].

### 3.2. Progesterone

Progesterone is another sex steroid hormone that is produced primarily in the corpus luteum and the placenta in females [17]. It is also synthesized in lower concentrations by the adrenal glands in both males and females [18]. In the menstrual cycle, progesterone increases the proliferation of the endothelial lining of the endometrium and thickens the endometrial wall [19]. This increased thickness and surface area of the endometrium plays a crucial role in the maintenance of the uterus during pregnancy [19]. In addition to its roles in the implantation and maintenance of pregnancy, progesterone has functions within the skeletal, cardiovascular, and nervous systems. For example, during the initiation of bone modeling, progesterone increases the process of bone formation [19]. Within the cardiovascular system, it plays an important role in regulating vascular tone and lowering blood pressure [20]. In the nervous system, progesterone is involved in the myelination processes and astroglial plasticity in addition to demonstrating neuroprotective effects and aiding in neuronal survival [19].

Progesterone may also affect metabolic pathways [21]. Menopausal hormonal therapy using micronized progesterone was found to either have no effects on fasting blood glucose and insulin resistance or to improve these parameters in non-diabetic postmenopausal women [22]. However, these effects can be attributed to estrogen or combined estrogen and progesterone, rather than the progesterone component alone [22]. In fact, progesterone has been implicated in insulin resistance during pregnancy [23], with the effect thought to be mediated by inhibition of the PI3K pathway as well as suppression of the PI3K-independent signaling pathways [23]. Administration of progestin (a synthetic form of progesterone) for contraceptive usage was also associated with an increased incidence of type 2 diabetes [24]. In animal studies, enhanced progression of diabetes in mice was associated with higher levels of circulating progesterone [25]. Additionally, female mice with knockout (targeted deletion) of progesterone receptors had a lower fasting blood glucose level and a higher concentration of circulating insulin compared to wild-type mice with intact progesterone receptors [25]. Progesterone may also impair the ability of the islet beta cells to secrete insulin, thus delaying glucose disposal and reducing glucose tolerance [26].

### 3.3. Luteinizing Hormone

Luteinizing hormone (LH) is a gonadotrophic (gonad-stimulating) hormone present in both males and females and is released by the anterior pituitary gland in response to the secretion of the hypothalamic gonadotropin-releasing hormone [27]. In females, LH stimulates the secretion of steroid hormones from the ovaries and helps with ovulation and implantation [28]. The LH surge during the ovulatory phase is in fact what causes the release of the ovum from the ovary [27]. Following this event, LH triggers the corpus luteum to begin progesterone synthesis and secretion, resulting in the growth of the endometrium [27]. Outside of the reproductive system, LH may play a role in learning, memory, and cognitive functions [29].

### 3.4. Follicle Stimulating Hormone

Follicle stimulating hormone (FSH) is another gonadotrophic hormone that is present in both males and females and is secreted by the anterior pituitary gland in response to the hypothalamic gonadotropin-releasing hormone [30]. In females, FSH functions to induce ovarian follicular growth and development in addition to estrogen production [30]. Besides these reproductive roles, FSH may also affect glucose metabolism. A recent systematic review suggests an inverse relationship between FSH levels and insulin resistance and a beneficial effect of FSH on glucose metabolism in postmenopausal women [31]. One of the studies discussed by this systematic review examined the relationship between FSH level, type 2 diabetes, and insulin resistance in postmenopausal women and found that higher FSH levels were associated with lower prevalence and incidence of type 2 diabetes and lower fasting insulin levels in this population [32]. High FSH levels were also linked to lower hemoglobin A1c (HbA1c) and fasting plasma glucose levels in postmenopausal women [33]. In addition, higher baseline FSH levels were associated with greater reductions in fasting glucose levels after six months of hormone therapy in midlife women (40 to 60 years of age) with menopausal symptoms [34]. However, some data do not support this trend: one animal study showed that high doses of FSH increased fasting glucose levels in mice, potentially through targeting the gluconeogenesis pathway [35], and another study reported no significant association between FSH and fasting plasma glucose levels in perimenopausal women [36].

## 4. Menstrual Cycle and Its Effects on Hormonal and Metabolic Responses to Exercise

The ovarian hormones exert significant metabolic effects in addition to regulating the female menstrual cycle, leading to variations in substrate metabolism between males and females in response to exercise [37]. For example, females rely less on carbohydrates and more on fat in response to the same relative exercise intensity compared to males [38]. In addition, in females varying levels of ovarian hormones throughout the menstrual cycle may affect substrate metabolism in response to exercise [37], although findings from early studies are controversial. In an early interventional study, glucose utilization and substrate oxidation were not significantly different between different phases of the menstrual cycle either at rest or during 90 min of moderate-intensity exercise (cycling at 50% VO_2_ max) in eumenorrheic females without diabetes (n = 13) [39]. Similarly, glucose turnover was not affected by the menstrual cycle phase during 60 min of ergometer cycling at 45% and 65% VO_2_ peaks in eumenorrheic women without diabetes (n = 8) [40].

Other studies, on the other hand, found lower rates of glucose appearance and disappearance, less carbohydrate oxidation, lower total glycogen utilization, and higher fat oxidation in the luteal compared to the follicular phase in response to moderate-intensity endurance exercise in eumenorrheic women without diabetes [41,42]. Conversely, apart from a significant difference in estrogen and progesterone levels between the follicular and luteal phases, no significant differences were observed between the two menstrual cycle phases with regards to the serum insulin, cortisol, growth hormone, glycerol, non-esterified fatty acids, lactate, or even lipid and carbohydrate oxidation rate and glucose utilization during maintained hyperglycemia (clamped blood glucose at 10 mM) throughout submaximal exercise (60% VO_2_ peak) [43]. These findings may suggest that metabolic effects resulting from ovarian hormone fluctuations in eumenorrheic females during moderate-intensity exercise may diminish in the presence of supraphysiological glucose levels, suggesting that the consumption of carbohydrates during endurance exercise may curtail metabolic variations caused by the menstrual cycle phase [43]. In the absence of carbohydrate consumption, however, it has been suggested that ovarian hormones may be capable of modifying exercise metabolism, as estrogen can act as a potent promoter of lipid oxidation [44]. In particular, this modification of metabolism might be the case when the estrogen-to-progesterone ratio is adequately elevated (e.g., when the magnitude of the increase in estrogen is at least twofold from the follicular to the luteal phase) [45].

Interestingly, oral contraceptives reduced glucose turnover rate without changing carbohydrate and lipid oxidation rates during 60 min of ergometer cycling at 45% and 65% VO_2_ peaks in eumenorrheic women without diabetes (n = 8) [46]. It was further suggested by the same group that although fluctuations of endogenous ovarian steroids may have little to no effect on exercise-induced lipolysis and triglyceride mobilization, synthetic ovarian steroids in oral contraceptives can increase triglyceride mobilization through increased lipolysis during moderate-intensity exercise in women [47]. Some support for this suggestion was provided by Isacco et al. (2014) [48], who found that oral contraceptives increased lipid mobilization when both rest and exercise (45 min of ergometer cycling at 65% VO_2_ max) sessions were pooled in the postprandial state in women without diabetes (n = 11). However, this effect was blunted on lipolytic activity stimulated by the exercise itself [48].

It has also been suggested that fed or fasting status has a greater impact on substrate oxidation in women than oral contraceptive use [49]. Isacco et al. (2012) [49] found that oral contraceptives modified neither fuel selection nor metabolic and hormonal responses during 45 min of exercise at 65% VO_2_ max in the fed state in women without diabetes (n = 21). Exercising while fasting, on the other hand, decreased carbohydrate oxidation during exercise, leading to higher lipid mobilization and utilization regardless of oral contraceptive status [49].

It is important to note that in terms of the effect of the menstrual cycle phase and hormonal fluctuations on substrate oxidation during exercise, some of the above-mentioned discrepancies may be partly due to the effect of exercise duration, individual variability in hormonal fluctuation, women’s nutritional status in influencing substrate metabolism and, most importantly, exercise intensity [38], as reported by a number of early studies before the 2000s. An interventional study on eumenorrheic women (n = 9) performing treadmill running at 35%, 60%, and 75% VO_2_ max showed reduced glucose metabolism at 35% and 60% VO_2_ max during the luteal phase compared to the follicular phase [50]. At 75% VO_2_ max, however, no menstrual cycle effects on glucose metabolism were observed [50], suggesting that at higher intensities, menstrual cycle phases may have little to no effect on substrate oxidation [50]. Similarly, an aerobic/treadmill exercise session at 70% VO_2_ max led to subtle differences in blood glucose response to exercise between menstrual cycle phases in eumenorrheic women without diabetes (n = 8) [51]. Interestingly, a high-intensity exercise (70% VO_2_ max) combined with a high carbohydrate diet (75% total caloric intake) was the most effective at minimizing the effects of the menstrual cycle on blood glucose response to exercise in this population (n = 9) [52].

As for muscle glycogen use, it seems that although there may not be a significant difference between menstrual cycle phases in response to aerobic exercise [53], anaerobic/high-intensity exercise may elicit a difference in muscle glycogen use between the two phases. One interventional study on physically active eumenorrheic women (n = 11) performing high-intensity intermittent exercise until exhaustion showed greater muscle glycogen use (measured by ^13^C-magnetic resonance spectroscopy) during the late follicular compared to the early follicular phase [54]. Estrogen levels and muscle glycogen use were found to be positively correlated, suggesting that estrogen increased reliance on glycogen at high intensities [54]. As for muscle glycogen repletion, however, it seems that there may not be a significant difference between menstrual cycle phases after exhaustive glycogen-depleting exercise in eumenorrheic women [55].

In addition to substrate metabolism, the effect of the menstrual cycle on exercise performance is another topic of debate. Some studies have shown no changes in exercise performance between different phases of the menstrual cycle, while others have reported improved performance during either early follicular, ovulatory, or mid-luteal phases [56]. Although a consensus is yet to be reached on whether exercise performance is influenced by the menstrual cycle phase, a recent meta-analysis of 51 studies indicated a trivial effect (median pooled effect size: −0.06) of reduced exercise performance during the early follicular phase compared to other phases in eumenorrheic women without diabetes [56], albeit with large variance among studies (τ_0.5_ = 0.26) and a large number of low-quality studies. This finding was attributed to the lower levels of estrogen and progesterone during the early follicular phase and their impact on muscular and overall exercise performance [56]. Similarly, another systematic review and meta-analysis of 21 studies showed non-significant, small, or trivial (effect sizes: *p* ≥ 0.26) differences between phases of the menstrual cycle for strength-related measures in eumenorrheic women without diabetes [57].

## 5. Menstrual Cycle and Type 1 Diabetes

Type 1 diabetes (T1D) is an autoimmune condition that involves the destruction of pancreatic beta cells, which are responsible for insulin production [58], resulting in a need for insulin replacement therapy [58]. Women with T1D are more likely to have menstrual cycle irregularities, generally consisting of alterations in the duration of the cycle, oligomenorrhea, and amenorrhea, especially with inadequate glycemic management [59,60,61]. Gaete et al. (2010) [59] found that increased cycle duration and oligomenorrhea were more frequent in T1D adolescents with higher HbA1c levels (7.6–8.9%) compared to controls without diabetes. In particular, for each point increase in HbA1c, the duration of the menstrual cycle increased by about 5 days [59]. This finding is consistent with retrospective data from Schroeder et al. (2000) [60], who found that menstrual irregularities increased significantly when HbA1c levels were above 10% in an adolescent T1D population. However, abnormalities of the menstrual cycle were observed even in T1D adolescents with optimal glycemic management, who exhibited twice the prevalence of oligomenorrhea than adolescents without diabetes [59].

These findings suggest that menstrual cycle irregularities are more prevalent in young women with T1D, with some evidence indicating that they may decrease with age. Strotmeyer et al. (2003) [61] found that although women with T1D had more menstrual problems, such as long cycles and long menstruation before the age of 30, than their sisters and unrelated control participants without diabetes, these differences were no longer evident after the age of 30. Additionally, cross-sectional data for adolescents with pre-pubertal T1D suggest that they experience later onset of menarche than both adolescents who developed T1D after menarche and control participants without diabetes. Menarche age, however, was not associated with HbA1c level in this T1D population [62]. Therefore, it seems that, compared to some of the above-mentioned menstrual irregularities in which HbA1c level may partly play a role, menarche is still delayed in T1D adolescents despite good glycemic management [63]. As such, it is likely that other factors besides average blood glucose levels are involved in producing menstrual irregularities in those with T1D.

Regardless of age, later menarche, earlier self-reported natural menopause, fewer pregnancies, and more stillbirths have been reported in women with T1D than those without diabetes across the reproductive lifespan [61]. Data from a prospective study suggest that this phenomenon is more pronounced when the onset of T1D occurs at an earlier age; a shorter reproductive period, i.e., delayed menarche and earlier menopause, has been reported in women with T1D onset prior to menarche compared to women without diabetes [64]. Conversely, when T1D developed after menarche, no association was found between T1D and age at menarche, age at natural menopause, or length of the reproductive years [64].

Women with T1D also have a higher frequency of hypogonadotropic hypogonadism, hyperandrogenism, and polycystic ovarian syndrome than those without diabetes, which can be linked to menstrual cycle irregularities [65]. For example, oligomenorrhoea, which is a common menstrual irregularity in T1D adolescents [66], may be due to hyperandrogenism in this population [67]. The disruption of the hypothalamus–pituitary–ovary axis and altered ovarian function caused by endogenous insulin deficiency, hyperglycemia, and exogenous hyperinsulinemia may partly explain some of the underlying causes of menstrual irregularities in T1D women [65]. Insulin plays an important role in maintaining the normal function of the hypothalamus–pituitary–ovary axis [65]. Altered ovarian function in T1D may be in part because of lower gonadotrophin levels due to decreased secretion of the gonadotropin-releasing hormone as a result of insulin deficiency [65]. On the other hand, exogenous hyperinsulinemia caused by intensive therapeutic protocols can potentially increase the exposure of the ovary to insulin. This exposure can facilitate androgen synthesis [68], which increases the risk of hyperandrogenism and polycystic ovarian syndrome in women with T1D [65].

Studies suggest that women with menstrual cycle irregularities, regardless of diabetes status, may be at increased risk of developing cardiovascular diseases [69,70,71]. For example, Wang et al. (2011) [69] found an increase in age-adjusted risk for coronary heart disease mortality (hazard ratio: 1.42) in women with irregular menstrual cycles during 456,298.5 person-years of follow-up. Similarly, Kiconco et al. (2022) [70] reported a 20% higher risk of developing heart disease in women with irregular menstrual cycles than those with regular menstrual cycles over a 20-year follow-up period (n = 13,714). Additionally, for every one-year increase in age, the risk increased by 11% [70]. Prospective data for 82,439 females also showed a 50% greater risk of non-fatal myocardial infarction or fatal coronary heart disease in women with very irregular menstrual cycles compared to those with a history of very regular menstrual cycles [71].

The reason behind such increased risk may be associated with the presence of polycystic ovarian syndrome in those with cycle irregularity and its link to metabolic abnormalities that predispose the person to cardiovascular diseases [71]. In addition, there is clinical evidence of hypoestrogenism in women with T1D, as well as lower serum estrogenic activity levels in adolescent females with T1D compared to those without diabetes [72,73]. This evidence may further explain the increased risk of cardiovascular diseases in this population, as cardioprotection in women during reproductive age is related, at least in part, to estrogens, especially estradiol [7]. Additionally, the use of oral contraceptives in adolescent females with T1D may be associated with a poorer cardiovascular risk profile [74], although further research is still required in this understudied area.

### 5.1. Menstrual Cycle, Blood Glucose, and Insulin Sensitivity

#### 5.1.1. Observational Studies in T1D Population

Cyclic changes in blood glucose levels and insulin sensitivity have been reported in a subset of women with T1D throughout the menstrual cycle [75,76,77,78,79]. In a retrospective study on women with T1D (n = 26), almost two-thirds of participants (65.4%) experienced cyclic changes in glycemia with an increase in blood glucose from the early follicular to the late luteal phase [79]. Similarly, Barata et al. (2013) [76] reported an increase in the percentage of hyperglycemia as well as a decrease in the percentage of hypoglycemia in the luteal compared to the follicular phase in each T1D participant (n = 6). This finding is consistent with a report of an increase in the frequency of hyperglycemia during the luteal phase in two out of four women with T1D [75], a pattern that was consistent over several cycles of the same participants [75]. Decreased time in range (euglycemia) during the mid and late luteal phase compared to the early follicular phase has also been found in 24 women with T1D, with time above range being significantly higher during the late luteal phase and time below range being significantly higher during the mid-follicular, compared to the early follicular phase [80]. The high frequency of hyperglycemia during the luteal phase was associated with lower insulin sensitivity in the early to mid and late luteal phase compared to the early follicular phase in women with T1D (n = 12) [78].

Conversely, some studies have reported no significant differences in mean glucose or insulin sensitivity among menstrual cycle phases [77,81]. Trout et al. (2007) [77], for example, found no significant differences in mean insulin sensitivity between the luteal and the follicular phase in T1D women using insulin pumps, although three out of five participants showed reduced insulin sensitivity during the luteal phase as measured with the Frequently Sampled Intravenous Glucose Tolerance Test. This study also reported increased non-insulin-mediated glucose disposal due to elevated blood glucose levels in the luteal phase [77]. Similarly, Levy et al. (2022) [81] found that insulin delivery and glycemic metrics remained stable throughout different phases of the menstrual cycle in 16 women with T1D using an automated insulin delivery system (t:slim X2 insulin pump with Control-IQ Technology and continuous glucose monitor). The authors suggested that this unexpected finding may be due to the mitigation of meaningful differences in glycemic outcomes as a result of more strategic, automated insulin delivery as opposed to reactive changes in insulin delivery in an open-loop setting [81].

#### 5.1.2. Observational Studies in Non-T1D Population

In women without diabetes, a longitudinal study of a large cohort of 257 women showed that insulin and measures of insulin resistance changed over the menstrual cycle, with both reaching maximum levels during the luteal phase [82]. These fluctuations were positively associated with estradiol and progesterone concentrations and negatively associated with FSH and sex hormone binding globulin [82], suggesting that changes in hormonal profile throughout the menstrual cycle may play a role in insulin sensitivity fluctuations in women. Cross-sectional data for 1906 women without diabetes showed cyclic changes in glucose but not in insulin or measures of insulin resistance [83]. After adjusting for body mass index, physical activity, and cardiorespiratory fitness, however, cyclic changes were observed not only for glucose but also for insulin and measures of insulin sensitivity in the study population [83]. This could yet again suggest a role for hormonal changes in insulin sensitivity fluctuation across the menstrual cycle, as body mass index and physical activity are both known to affect ovarian hormonal profiles [84,85].

### 5.2. The Menstrual Cycle, Exercise, and T1D

As discussed, although the evidence is inconclusive, the menstrual cycle can affect substrate metabolism differently depending on the cycle phase. Additionally, the menstrual cycle can affect hormonal and metabolic responses to exercise differently depending on the cycle phase, as well as the type and intensity of exercise. However, the effects of both the menstrual cycle and exercise on substrate metabolism have not been extensively studied in females with T1D.

For individuals without T1D, variable levels of glucose and/or insulin sensitivity during the menstrual cycle (if encountered) may not have noticeable or adverse effects. For those with T1D, however, complications can emerge as a result of the menstrual cycle. One such problem arises from decreased insulin sensitivity or higher blood glucose levels, which are often experienced by a subset of women with T1D during the luteal phase, as discussed earlier. Higher glucose levels during the luteal phase could potentially protect against hypoglycemia both during and after exercise in those with T1D. The relatively higher estrogen levels during the luteal phase have been associated with a higher reliance on lipids as a fuel source during exercise [86]. Additionally, the luteal phase has been associated with less glycogen usage during exercise [42]. Combined, these factors may offer protection against hypoglycemia during and after exercise in those with T1D when exercise takes place during the luteal phase of the menstrual cycle.

However, to counteract hyperglycemia and maintain healthy blood glucose levels, women with T1D tend to increase their insulin doses when they experience reduced insulin sensitivity or high blood glucose during the luteal phase [87]. Depending on the type of insulin and the timing of administration, this adjustment could increase the risk of hypoglycemia both during and after exercise due to higher levels of insulin in circulation during exercise (Figure 2). The occurrence and fear of hypoglycemia is a significant barrier to exercise for those with T1D [88], especially for females with T1D who tend to consume less carbohydrates than T1D males to prevent activity-related hypoglycemia [89]. This fear may partly explain why females with T1D are less active than their male counterparts [89,90] despite the many health benefits of exercise and physical activity for this population [91]. It is therefore essential that the effects of the menstrual cycle on blood glucose management be assessed further during both rest and exercise in those with T1D in order to alleviate such fear and its associated negative consequences.

Very few studies have been conducted to understand the effect of the menstrual cycle on blood glucose response to exercise in females with T1D. McGaugh et al. (2020) [92] examined glycemic response to prolonged (2 h) aerobic exercise (~45% VO_2_ peak) while fasting in eumenorrheic females with T1D (n = 7) during the luteal and early-follicular phases of the menstrual cycle. No significant differences in glucose levels, carbohydrate consumption, or carbohydrate oxidation were found between the two phases before and during exercise [92]. However, post-exercise nocturnal hyperglycemia was more frequent during the luteal compared to the early follicular phase [92]. On the other hand, Momeni et al. (2022) [93] found a higher frequency of post-exercise hyperglycemia during the early follicular compared to the late luteal phase in eumenorrheic females with T1D (n = 9) performing non-fasting-state moderate aerobic exercise at 50% VO_2_ peak. Additionally, more glucose supplementation was required during the late luteal phase to prevent hypoglycemia during and after exercise [93]. This observed hypoglycemia could be due to more insulin in circulation as a result of reduced insulin sensitivity and/or higher glucose levels during this phase of the menstrual cycle, as previously reported [75,76,77,78,79].

To date, no other studies have investigated the effects of acute and chronic exercise interventions on menstrual cycle-related insulin sensitivity and glucose response in females with T1D. Other studies on non-T1D populations, such as those with prediabetes or type 2 diabetes or those without diabetes, have shown that regular exercise and physical activity, especially when in line with the guidelines, can improve insulin sensitivity [94,95]. Although not unanimous, some studies report a dose–response relationship, where higher exercise volume and intensity produces greater benefits in insulin sensitivity [94]. The combination of aerobic and resistance exercise training has also been suggested to be more effective than either mode alone [94].

In addition to insulin sensitivity, exercise can affect insulin clearance and glucose sensitivity. In particular, both exercise mode and sex affect glucose sensitivity and insulin clearance in those without diabetes [96]. Following an acute bout of moderate-intensity exercise, for example, insulin clearance is higher in females than males. Interestingly, in females insulin clearance and glucose sensitivity are greater following high-intensity interval exercise compared to moderate-intensity continuous exercise [96]. As for T1D, however, more research is required to examine these findings, especially with regard to the menstrual cycle and the effects of different cycle phases.

## 6. Conclusions

Many of the studies involving the menstrual cycle and T1D have a small sample size and report changes in glycemia and insulin sensitivity only in a subset of women with T1D with high intra- and inter-subject variabilities. Nonetheless, these findings can emphasize the importance of considering the phases of the menstrual cycle in order to predict the associated glycemic risks as a way to plan appropriate insulin therapy. As for exercise studies involving the effects of the menstrual cycle in those with T1D, a great deal more research is required to understand glycemic response to exercise in this population. The new evidence can either confirm that the existing insulin adjustment and carbohydrate intake guidelines are appropriate for females with T1D in different phases of the menstrual cycle, or can help develop new sex-specific recommendations based on the menstrual cycle phase. These outcomes could eliminate an important barrier to exercise in females with T1D, which has the potential to increase activity, improve mental health and quality of life, and decrease the risk of diabetes-related complications in this population. In addition, as medical engineers move forward in developing closed-loop insulin delivery systems, the new data will help to better manage blood glucose during exercise in the context of menstrual-cycle fluctuations in insulin sensitivity in females with T1D.

## Figures and Tables

**Figure 1 ijerph-20-02772-f001:**
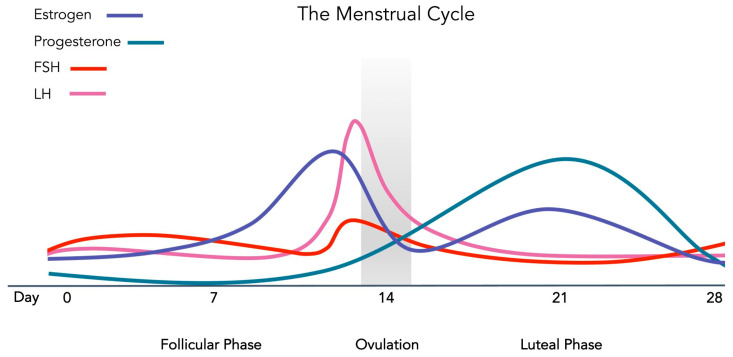
Phases and hormonal profile of the menstrual cycle.

**Figure 2 ijerph-20-02772-f002:**
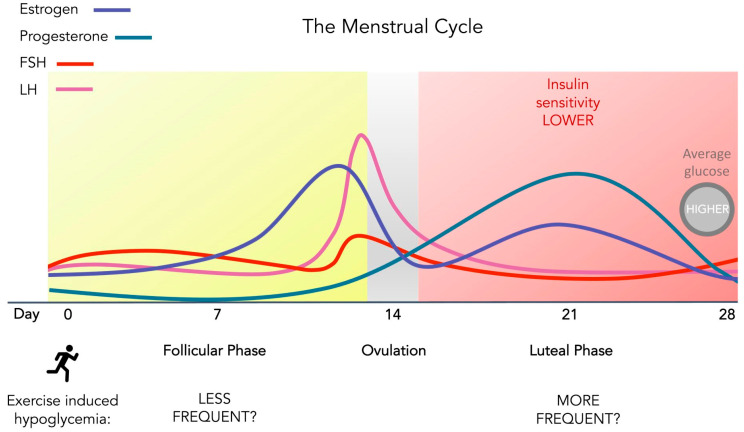
Variable levels of glucose and/or insulin sensitivity during the menstrual cycle in females with T1D. Lower insulin sensitivity and/or higher blood glucose levels are often experienced by a subset of females with T1D during the luteal phase. The effects that these fluctuations may have on blood glucose levels during and after exercise are not yet clear in this population.

## Data Availability

Not applicable.

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
