# Peer review of "Type 1 Diabetes and the Menstrual Cycle: Where/How Does Exercise Fit in?"

_ijerph, 2023, doi:10.3390/ijerph20042772_

Round 1

Reviewer 1 Report

This is an interesting and clearly written review. Authors have explained how menstrual hormones in women with type 1 diabetes can impact exercise performance. However, the authors can add a brief note on signaling pathways that connect menstrual hormone production and their effect on insulin production and insulin sensitivity. A schematic diagram depicting these pathways would benefit the readers. 

Author Response

We would like to thank the reviewer for taking the time to read our manuscript and provide feedback. We have done our best to address the comments and concerns raised. Below, please find responses to reviewer comments. The resulting changes have been made using the track changes function in the revised manuscript.

Authors’ response: Thank you for your comment.

We did initially try to add some of the signaling pathways that may be potentially involved in insulin sensitivity changes during the menstrual cycle. We have realized that the focus of the current literature has been mostly on estrogen (and to a lesser extent on progesterone) and there is a lack of information regarding other hormones. As for the estrogen itself, different signaling pathways/molecules have been examined, with some reporting controversial findings.

Many of the studies have been performed in the context of type 2 diabetes and insulin resistance, and others on a wide range of other animal/cell models, which may not be necessarily applicable in the context of type 1 diabetes. In order to discuss these pathways, we needed to introduce and explain a wide range of upstream and downstream molecules and pathways which would take the focus away from the main objectives of the review which is for clinical purposes. We, therefore, decided to avoid complications by not introducing these pathways in a schematic diagram but to still address the first part of your recommendation, i.e., a brief note on the signaling pathways.

We have added the following studies (that seem to have more consistent findings) to the estrogen and progesterone sections of the paper:

(page 3, line 105) “It is suggested that GPER (G Protein-Coupled Estrogen Receptor GPR30) activation of the epidermal growth factor receptor and ERK (extracellular signal-regulated kinase) in response to estradiol may play an important role in the secretion of insulin from beta-cells [12].”

(page 3, line 110) “Estradiol effects on insulin sensitivity are suggested to be mediated through the inhibition of transcription factor Foxo1 via activation of ERα-PI3K-Akt (estrogen receptor alpha-phosphoinositide 3 kinases-protein kinase B) signaling pathway [14].’

(page 4, line 141) “Progesterone has, in fact, been implicated in insulin resistance during pregnancy [24], with the effect thought to be mediated by inhibition of the PI3K pathway as well as suppression of the PI3K-independent signaling pathways [24].”

Reviewer 2 Report

PLEASE FIND ATTACHED

Author Response

We would like to thank the reviewer for taking the time to read our manuscript and provide feedback. We appreciate your comments and are glad that enjoyed our work. 

Reviewer 3 Report

GENERAL COMMENTS

Paper addresses an important aspect of health, which is assessing the effect of the menstrual cycle on blood glucose responses to exercise in females with Type 1 Diabetes.

Several studies have explored different aspects of Diabetes. This paper is an addition with context relevance for Primary Health Care, hence, I consider it appropriate for publication. However, additional revision and proof reading is recommended to improve the content, formatting, and language of the paper.

The abstract is descriptive, but not very well structured, and the objectives of the study were not clearly stated to justify the reason for which the study is relevant.

Findings are specific in describing the effect of exercise on Type 1 diabetes and the menstrual cycle, and the conclusion adequately aligns with findings.

The background was well written and the need for the study justified, but I am of the view that it should be summarised to reduce the content and make it more simplified.

The objectives of the study were however not explicitly stated.

The methodological orientation of the study was not well described, the Authors did not specify the number of studies they reviewed, as well as the methodology used for the review.

Author Response

We would like to thank the reviewer for taking the time to read our manuscript and provide feedback. We have done our best to address the comments and concerns raised. Attached, please find responses to reviewer comments. The resulting changes have been made using the track changes function in the revised manuscript.
